# ReTa-Diffusion: Exploring Task-State from Resting-State in EEG Signals via Bidirectional Decoupling and Latent Guiding for Early Detection of Subclinical Depression

## Abstract

Depression is a persistent and difficult-to-treat condition that poses a serious threat to public health. Early detection of subclinical or subthreshold depression (SD) is critical for preventing its progression into major depressive disorder. Compared to other modalities like wrist-worn wearables, electroencephalography (EEG) offers greater clinical value due to its direct link to brain function. Task-state EEG can reflect the characteristics of task related brain regions, its requirement for subjects to perform time-consuming tasks renders it impractical for rapid, large-scale screening. Whereas, resting-state EEG (rs-EEG) has advantages of task-free nature and ease of acquisition. Crucially, it contains spontaneous neural activity and can reflect the dynamic information across the entire brain, which is theorized to contain the neural substrates of all potential task-states. This makes it a highly valuable, yet underutilized, tool for SD prediction and diagnosis. However, effectively extracting task-specific information from rs-EEG remains a major challenge, hindered by two primary issues: (1) the limited understanding of relationships between resting-state and task-state EEG, and (2) the absence of effective guidance for feature extraction of generated task related EEG. To address the issues, we propose ReTa-Diffusion, a novel framework designed to mine task-related features from rs-EEG for the early detection of SD. It comprises two core modules: a Bidirectional Decoupled Conditional Diffusion (BDCD) module and a Wavelet-Riemannian Feature Extraction (WRFE) module. Inspired by the principle that task-state EEG can guide the interpretation of resting-state data to improve diagnostic accuracy, the BDCD addresses issue (1) by disentangling and aligning features from resting and task states through a bidirectional diffusion mechanism, thereby generating task-state-informed EEG signals from resting-state data. Subsequently, the WRFE tackles issue (2) by capturing rich dynamic temporal patterns and functional connectivity from these generated signals, leveraging Riemannian manifolds and a cross-attention mechanism. Together, these modules enable effective feature learning from rs-EEG, significantly enhancing the generalizability and accuracy of early SD detection. The proposed ReTa-Diffusion is evaluated using five-fold cross-validation on three datasets, achieving a classification accuracy of 54.52% on multiclass classification tasks, outperforming existing state-of-the-art methods by 18.95%. Further validation through leave-one-subject-out and visual analysis confirms its robust cross-subject capability and the effectiveness of using task-state features to guide stress-level prediction from rs-EEG. These results underscore ReTa-Diffusion's potential as a powerful tool for early depression screening and prevention.

## 1 Introduction

In recent years, depression has emerged a major global public health concern. Subclinical/ Subthreshold depression (SD), regarded as an early and potentially reversible stage of the disorder, can greatly benefit from timely detection to prevent its progression (Cuijpers et al., 2004). Existing

studies predominantly rely on questionnaire-based assessments to evaluate SD, such as the Hamilton Depression Rating Scale (Snaith et al., 1996) and the Perceived Stress Scale (PSS) (Cohen et al., 1983), which are prone to response bias and subjective results, and also time consuming, resulting in poor effectiveness of early detection and intervention. Although current real-time monitoring systems, such as wristbands (Al-Alim et al., 2024) and bracelets (Zhong et al., 2022) using electrocardiograms (ECGs) and pulse signals, can effectively track psychological/mental stress, they are not well-suited for detecting SD (Jiang et al., 2019). Electroencephalogram (EEG) is among the most common modalities for mental stress measurement due to their noninvasive configuration, low cost, high mobility, and high temporal resolution (Elmousalami et al., 2025; Acharya et al., 2025). EEG can be obtained using either resting or task related experiments, i.e., resting-state EEG (rs-EEG), task-state EEG. Comparably, the rs-EEG offers richer insights into mental activity and is readily accessible, whereas task-state EEG, though more time-consuming, provides significant, task-related information about mental function. Building on the insights, we raise a question, would it be a better choice to employ only rs-EEG for clinical detection purposes?

Ghiasi et al (Ghiasi et al., 2021) used features of spectral and functional connectivity derived from rs-EEG to improve detection ability for depression. To the best of our knowledge, there has been limited research using rs-EEG to predict subjects' stress levels state, due to the following issue, i.e., the underlying mapping between rs-EEG and task-state EEG, making it difficult to effectively extract task-related information from rs-EEG, and (2) the absence of effective guidance for feature extraction from the generated task-state EEG often leads to unstable and non-specific features, thereby limiting prediction performance.

To deal with the issue, we draw inspiration from energy transformation processes observed in

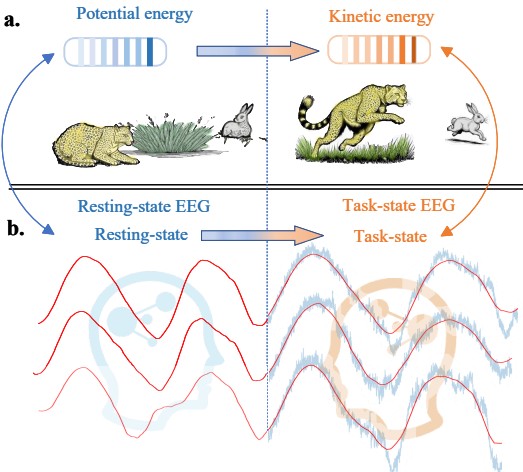

Figure 1: Nature-inspired analogy between energy states and EEG dynamics. (a) A leopard crouching in the grass appears still, yet its body is tense and alert—ready to strike. This reflects a shift from potential to kinetic energy. (b) Similarly, we consider rs-EEG as a latent preparatory state.

nature, as illustrated in Figure 1. The state of a leopard poised to strike upon detecting its prey (potential energy) can be considered as the rs-EEG, whereas the moment the leopard launches into pursuit (kinetic energy) corresponds to the task-state EEG. Based on this analogy, we propose using task-state EEG to guide the extraction of latent features from rs-EEG, i.e., decoupling specific task-state related EEG signals. To accurately predict the state/stress level and evaluate the generative model, effective and sufficient task related EEG features are needed in order to solve issue (2).

Therefore, we propose a novel framework termed ReTa-Diffusion to detect SD, that performs Resting-state to Task-state EEG generation and specific task-state prediction using Diffusion-based model. It includes two modules: (1) a Bidirectional Decoupled Conditional Diffusion (BDCD) module, which integrates a bidirectional decoupling mechanism into a diffusion model to disentangle rs-EEG and task-state EEG features, while uncovering their latent relationships to generate task-state EEG signals; (2) a Wavelet-Riemann Feature Extraction (WRFE) module, which combines wavelet convolution with Riemannian manifold to capture dynamic temporal features and functional connectivity from the generate task-state EEG signals to predict current stress level. The main contributions

of this paper are as follows:

(1) Unlike existing studies that focuses on evaluating SD using both rs-EEG and task-state EEG, to our best knowledge, we are the first to emphasize the importance of identifying latent task-related neural activity within rs-EEG and the leave-one-subject-out validation is used to evaluate the model's cross-subject capability, providing new perspectives and approaches for early detection of SD.

(2) On the basis of contribution (1), we propose a novel diffusion-based detection framework named ReTa-Diffusion consisting of the BDCD and WRFE modules. The BDCD module decouples rs-EEG and task-state EEG features to guide the diffusion model in effectively capturing latent task-related neural patterns embedded in the resting states.

(3) Building on the task-state EEG signals generated by the BDCD module, the WRFE module combines the wavelet convolution with Riemannian manifold, and spatial features in the generated task-state EEG. Unlike existing work, we focus on both temporal dynamics and intrinsic geometric structure from signals, therefore extracting more sufficient and effective features.

## 2  METHODOLOGY

In this section, we introduce the proposed multiclass classification framework, termed ReTa-Diffusion. The overall architecture, as illustrated in Fig. 2, consists of three sequential phases: signal preprocessing, a BDCD (Bidirectional Decoupling and Cross-domain Diffusion) module, and a WRFE (Wavelet-Riemannian Feature Extraction) module. The signal preprocessing phase provides clean and reliable EEG signals. The BDCD module then extracts latent features from these signals to generate high-fidelity, task-state-related EEG representations. Finally, the WRFE module extracts discriminative temporal and spatial features from the generated signals to produce the final stress level classification.

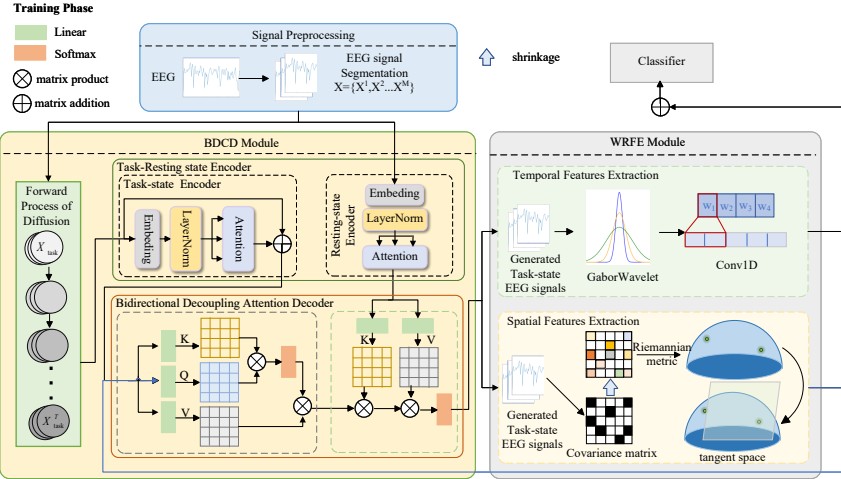

Figure 2: Framework of ReTa-Diffusion. It consists of three stages: (1) Signal Preprocessing for segmenting EEG signals into slices; (2) the BDCD based on Diffusion captures the dynamic correlation between resting-state and task-state EEG signals to generate task-state EEG signals; (3) the WRFE extracts both the temporal and spatial features of EEG signals for optimising the Decoder

### 2.1  DATA PREPROCESSING

Raw EEG signals are initially filtered using a 50 Hz lowpass filter to remove power-line interference and high-frequency noise. The filtered EEG signals are then down sampled to 250 Hz, retaining the main neural oscillation bands in EEG signals while reducing data dimensionality and improving computational efficiency. Next, Independent Component Analysis (ICA) is employed to remove artifacts. Finally, when creating the sample, we slice the continuous EEG signals $S_L^U$ (where $U$ denotes the number of subjects and $L$ is the signal length), are segmented into overlapping epochs. Each epoch is denoted as $X = \{X_{\text{task}}, X_{\text{rest}}\} \in \mathbb{R}^{d \times l}$ , where $d$ represents the number of chan-

nels, and $l$ denotes the duration of each epoch. Our experimental results demonstrate that optimal performance is achieved when $l = 1$ is set to second(s).

## 2.2 EEG GENERATION BY BDCD MODULE

This module on the basis of diffusion model consists of three components: a task-state EEG signal noising process, a Task-Resting Encoder, and a Bidirectional Decoupling Attention Decoder. First, noise is progressively added to the sliced task-state EEG signals through a forward diffusion process. The noised signals, along with the corresponding resting-state EEG signals, are then fed into the Task-Resting Encoder to extract domain-invariant latent features. Finally, these latent features are calibrated and refined by the Bidirectional Decoupling Attention Decoder, and the output is passed through a Multi-Layer Perceptron (MLP) to reconstruct the task-state EEG signals.

### 2.2.1 FORWARD DIFFUSION PROCESS

In the forward process of diffusion, Gaussian noise is iteratively added to the clean task-state EEG signals $X_{\text{task}}$ over $T$ time steps, until they asymptotically converge to an isotropic Gaussian distribution, denoted as $X_{\text{task}}^T \in \mathbb{R}^{d \times l}$. This process is formulated as a Markov chain:

$$X_{\text{task}}^\top = \sqrt{\overline{\alpha_t}}\, X_{\text{task}} + \sqrt{1 - \overline{\alpha_t}}\, \epsilon \tag{1}$$

where $\alpha_t = 1 - \beta_t$ and $\overline{\alpha}_t = \prod_{i=1}^T \alpha_i$, The noise schedule $\beta_t \in (0, 1)$ is a series of scalars gradually increasing over the time step $t$ satisfying $0 < \beta_1 < \beta_2 < \cdots < \beta_T < 1$. Based on established practices and our empirical analysis, we set the total number of time steps to $T = 1000$.

### 2.2.2 TASK-RESTING ENCODER

Task-Resting Encoder includes a Task-state Encoder and a Resting-state Encoder, which are designed to map $X_{\text{task}}^\top \in \mathbb{R}^{d \times l}$ and rs-EEG $X_{\text{rest}} \in \mathbb{R}^{d \times l}$ into a shared space, disentangling subject-specific noise while preserving task-related components that can be transferred different subjects, thereby effectively mitigating inter-subject variability. This process is represented as follows.First, adaptive layer normalization is applied to both inputs, conditioned on the time step t:

$$\overline{X_{\text{task}}^\top} = \frac{X_{\text{task}}^\top - \mu(X_{\text{task}}^\top)}{\sigma(X_{\text{task}}^\top)} \cdot \gamma(t) + \rho(t), \quad \overline{X_{\text{rest}}} = \frac{X_{\text{rest}} - \mu(X_{\text{rest}})}{\sigma(X_{\text{rest}})} \cdot \gamma(t) + \rho(t) \tag{2}$$

where $\sigma(t)$ is scaling parameters determined by the time step $t$ and $\rho(t)$ is the shifting parameters determined by the time step $t$. Next, a self-attention mechanism is applied to capture long-range dependencies within each signal. The attention weights are computed as:

$$A_{\text{task}}^i = \frac{\exp(\overline{X_{\text{task}}^\top} a_i)}{\sum_{i=1}^l \exp(\overline{X_{\text{task}}^\top} a_i)}, \quad A_{\text{rest}}^i = \frac{\exp(\overline{X_{\text{rest}}} a_i)}{\sum_{i=1}^l \exp(\overline{X_{\text{rest}}} a_i)} \tag{3}$$

where $a_i$ denotes the $i^{th}$ attention weight vector. $A_{\text{task}}^i \in \mathbb{R}^{l \times l}$ and $A_{\text{rest}}^i \in \mathbb{R}^{l \times l}$ are the resulting attention matrices. These attention matrices are then used to compute weighted sums of the normalized features, yielding the final latent representations for the shared space $h_{\text{task}}$ and $h_{\text{rest}}$.

$$h_{\text{task}} = \sum_{i=1}^l A_{\text{task}}^i \overline{X_{\text{task}}^\top}, \quad h_{\text{rest}} = \sum_{i=1}^l A_{\text{rest}}^i \overline{X_{\text{rest}}} \tag{4}$$

### 2.2.3 BIDIRECTIONAL DECOUPLING ATTENTION DECODER

The Bidirectional Decoupling Attention Decoder employs Multi-Head Attention (MHA) and Cross-Attention layers to refine the latent features. Initially, the task-state features $h_{\text{task}}$ are fed into the MHA module to capture complex intra-pattern dependencies, resulting in a refined representation $H_{\text{task}}$. MHA is selected for its proven ability to model diverse temporal relationships simultaneously. The MHA process is defined as:

$$Q_{\text{task}} = h_{\text{task}} W_Q, \quad K_{\text{task}} = h_{\text{task}} W_K, \quad V_{\text{task}} = h_{\text{task}} W_V \tag{5}$$

$$A_{\text{task}} = \frac{\exp\left(\frac{Q_{\text{task}} K_{\text{task}}}{\sqrt{d}}\right)}{\sum_{m=1}^{l} \exp\left(\frac{Q_{\text{task}} K_{\text{task}}}{\sqrt{d}}\right)} \tag{6}$$

$$H_{\text{task}} = \sum_{i=1}^{l} A_{\text{task}} V_{\text{task}} \tag{7}$$

where $W_Q, W_K, W_V \in \mathbb{R}^{d \times m}$ represent a learnable projection matrixs for for the Query, Key, and Value respectively. Subsequently, a cross-attention layer is used to calibrate the resting-state features $h_{\text{rest}}$ using the information from the refined task-state feature $H_{\text{task}}$. This allows the model to extract task-relevant information from the resting-state data. The process is illustrated below:

$$Q = H_{\text{task}} W_Q, \quad K_{\text{rest}} = h_{\text{rest}} W_K, \quad V_{\text{rest}} = h_{\text{rest}} W_V \tag{8}$$

$$A_{\text{cal}} = \frac{\exp\left(\frac{Q K_{\text{rest}}}{\sqrt{d}}\right)}{\sum_{m=1}^{l} \exp\left(\frac{Q K_{\text{rest}}}{\sqrt{d}}\right)} \tag{9}$$

$$H_{\text{cal}} = \sum_{i=1}^{l} A_{\text{cal}} V_{\text{rest}} \tag{10}$$

where $A_{\text{cal}}$ represents the calibrated attention weights and $H_{\text{cal}}$ denotes the calibrated features, which encode the task-relevant information extracted from the final reconstructed rs-EEG signals.

Finally, $H_{\text{cal}}$ is then passed through the MLP layer to generate the task-state EEG signal $\hat{X}_{\text{task}} \in \mathbb{R}^{d \times l}$.

## 2.3 SD DETECTION BY WRFE MODULE

This module consists of two components: Temporal Feature Extraction by Wavelet Convolution and Spatial Feature Extraction by Riemann. This module employs a dual-branch architecture to separately extract the temporal and spatial features of the generated EEG signals, and then fuses these features through a fully connected layer to produce the stress level classification.

### 2.3.1 TEMPORAL FEATURE EXTRACTION BY WAVELET CONVOLUTION

Compared with fixed-shape conventional convolution kernels, adaptive wavelet convolution can adjust the filter shape in the time domain, enabling more effective capture of the multi-band characteristics of the generated EEG signals $\hat{X}_{\text{task}}$. Since Gabor wavelets can adaptively capture the multi-band features of EEG signals, we choose Gabor as the convolution kernel, which is defined as shown in Eq. (11):

$$\psi_f(u; f, \omega) = \exp\left(-\frac{u^2}{2\,\omega^2}\right) \cdot \exp(2\pi i f u) \tag{11}$$

where $f$ is the center frequency, $\omega$ controls the temporal width, which are learnable parameters. Then, the generated task-state EEG signal $\hat{X}_{\text{task}}$ is fed into the Gabor wavelet convolution to effectively capture neural activities across different frequency bands, thereby extracting the temporal features of the EEG signal as follows:

$$G_{\text{temp}} = \text{conv}(\hat{X}_{\text{task}}, \psi_f) = \int_{-\infty}^{+\infty} \hat{X}_{\text{task}}\, \psi_f(u - \tau)\, du \tag{12}$$

The temporal feature $G_{\text{temp}}$ effects patterns of brain activity in EEG signals, where $u - \tau$ represents the position of the convolution kernel sliding over the signal.

### 2.3.2 SPATIAL FEATURE EXTRACTION BY RIEMANNIAN

Since processing in Euclidean space can disrupt the geometric relationships between channels in the generated EEG signals $\hat{X}_{\text{task}}$, we map $\hat{X}_{\text{task}}$ onto a Riemannian manifold to preserve their spatial structure. Firstly, the generated task-state EEG signal $\hat{X}_{\text{task}}$ is used to compute its covariance using

the covariance pooling layer, the covariance matrix $C_{i=1}^M \in \mathbb{R}^{d \times d}$ to obtain the relationship between EEG channels, as shown in Eq. (13):

$$C_{i=1}^M = \frac{1}{l-1} \hat{X}_{\text{task}} \hat{X}_{\text{task}}^T \tag{13}$$

Generally, the covariance matrix of EEG signals should be symmetric positive definite. However, due to the influence of noise, the estimated covariance matrix is often only positive semidefinite. Therefore, it is necessary to apply shrinkage to the covariance matrix, as presented in Eq. (14):

$$C_{\text{SPD}} = (1-\lambda)C_{i=1}^M + \lambda I \tag{14}$$

where $\lambda$ is provided by the unbiased estimator proposed by Ledoit (Ledoit et al. 2004), which can automatically determine the optimal value of $\lambda$. Since EEG signals are affected by electrode placement and scalp conductivity, which can introduce linear shifts, the affine-invariant Riemannian metric is used to compute the reference center $C_{\text{ref}}$, effectively resisting signal distortions caused by differences in experimental equipment and individual subjects, as shown in Eq. (15):

$$C_{\text{ref}} = \left\| \log \left( (C_{\text{SPD}}^i)^{-1/2} C_{\text{SPD}}^j (C_{\text{SPD}}^i)^{-1/2} \right) \right\|_F \tag{15}$$

We obtain reference points $C_{\text{ref}}$ of the generated EEG on the manifold. To preserve the geometric structure of the EEG data, the feature points on the manifold are mapped to the tangent space based on these reference points, as shown in Eq. (15):

$$S = C_{\text{ref}}^{1/2} \log \left( C_{\text{ref}}^{-1/2} C_{\text{SPD}} C_{\text{ref}}^{-1/2} \right) C_{\text{ref}}^{1/2} \tag{16}$$

Since the tangent space feature $S$ is a symmetric matrix, to remove redundancy, only its upper triangular part is taken and then unfolded into a feature vector, as shown in Eq. (16):

$$G_{\text{spat}} = \text{vec}(S) \tag{17}$$

where $G_{\text{spat}}$ represents the Riemannian manifold feature. Finally, The fused temporal and spatial features are fed into a fully connected layer to produce the final prediction $P$.

## 2.4 Loss Function

Following the methodology outlined by Benchetrit et al. (Benchetrit et al, 2018), we adopt a dual approach to loss functions $L_{recon}$ (in Eq. C1), between the original and reconstructed signals, and design a multi-dimensional sliding window covariance loss, $L_{cov}$ (in Eq. C2), to ensure statistical similarity. For the stress classification task, we employ the standard cross-entropy loss, $L_{cro}$ (in Eq. C3). Therefore, the overall loss function of our model is a weighted combination of these components, where is a hyperparameter used to balance the contribution of the generative and discriminative losses.

$$\text{Loss} = \lambda \left( L_{\text{recon}} + L_{\text{cov}} \right) + (1-\lambda)L_{\text{cro}} \tag{18}$$

where $\lambda = 0.5$ is used to balance the contribution of each loss.
The training and testing pseudo-code of our ReTa-Diffusion is presented in Appendix B.

## 3 Experimental Results

In this section, we compare our method with state-of-the-art (SOTA) methods and demonstrate its effectiveness through ablation studies and visualization experiments using three datasets (two private and one publicly available datasets (Wang et al., 2022)). A cross-subjects design was used for all experiments, with the aim of efficiently assessing the generalization ability of the proposed model.

## 3.1 Datasets

Three datasets are employed to evaluate the performance of ReTa-Diffusion for metal stress level detection, where both MIST-CNU and MIST-UTP are private datasets, and Rest-Task dataset (Wang et al. 2022) is a publicly available dataset. More details on datasets, System Environment and Parameter Configuration and Evaluation Metrics can be found in Appendix E.

Table 1: Performance comparison on MIST-CNU Dataset

| Method | ACC (%) | Pre (%) | AUC (%) | F1 (%) | Rec (%) |
|--------|---------|---------|---------|--------|---------|
| EEGNet | $35.57 \pm 3.96$ | $35.58 \pm 3.78$ | $56.23 \pm 3.97$ | $29.36 \pm 2.54$ | $35.57 \pm 3.95$ |
| MuLHiTA | $32.62 \pm 1.04$ | $29.75 \pm 0.23$ | $47.10 \pm 1.19$ | $28.77 \pm 0.23$ | $32.63 \pm 1.04$ |
| BIOT | $26.31 \pm 0.47$ | $34.45 \pm 0.01$ | $57.24 \pm 0.01$ | $22.46 \pm 0.04$ | $35.08 \pm 0.01$ |
| LaBraM | $34.70 \pm 0.10$ | $35.50 \pm 1.77$ | $42.46 \pm 1.47$ | $17.80 \pm 0.23$ | $34.56 \pm 0.78$ |
| **WRFE** | $\mathbf{54.52 \pm 0.13}$ | $\mathbf{67.01 \pm 0.14}$ | $\mathbf{83.12 \pm 0.04}$ | $\mathbf{51.12 \pm 0.04}$ | $\mathbf{54.52 \pm 0.01}$ |

## 3.2 COMPARISON WITH SOTA

Given the novelty of our proposed ReTa-Diffusion framework, a direct comparison with end-to-end counterparts is not feasible. Therefore, to validate the effectiveness of our design, we conduct separate evaluations for its two core components: the BDCD and the WRFE, comparing them against relevant SOTA baseline. To evaluate the cross-subject generalization capability of the complete ReTa-Diffusion framework, we conducted a case study on the MIST-CNU dataset using a leave-one-subject-out validation strategy. The process can be found in Appendix D.

### 3.2.1 COMPARISION BETWEEN BDCD MODULE AND SOTA

To assess the generalizability and effectiveness of the BDCD module in generating high-fidelity task-state EEG from resting-state data, we compare it against three prominent generative frameworks: C-WGAN (Sharma et al., 2018), TimeGAN (Yoon et al., 2019), and Diffusion-TS (Yuan et al., 2024). The quantitative results, evaluated using Sliced Wasserstein Distance (SWD) and Context-FID (lower is better), are presented in Table 1. As shown in Table 1, our BDCD module achieves superior performance across all evaluation metrics on the three datasets. Notably, BDCD significantly outperforms C-WGAN in terms of Context-FID, with reductions of 10, 30, and 17 on the MIST-CNU, MIST-UTP, and Rest-Task datasets, respectively. Furthermore, BDCD achieves a lower SWD (0.0768) compared to TimeGAN (0.0977) on the Rest-Task dataset. The most direct comparison is with Diffusion-TS, which serves as the backbone for our module. BDCD consistently surpasses Diffusion-TS, achieving Context-FID values that are 0.5 lower for MIST-CNU and MIST-UTP, and 0.3 lower for Rest-Task. These improvements can be attributed to the proposed bidirectional decoupling attention decoder, which effectively leverages task-state information to guide the generation process.

### 3.2.2 COMPARISION BETWEEN WRFE MODULE AND SOTA

The results of a comparison experiment are provided in Table 1, where it is evident that WRFE module outperformed other frameworks across all evaluation metrics. Specially, WRFE module outperformed the EEGNet (Lawhern et al., 2018) across MIST-CNU datasets, as measured by classification accuracy. Regarding LaBraM (Jiang et al., 2024), its performance is comparatively low, with an accuracy of 34.7%, precision of 35.5%, F1 of 17.8%, recall of 34.56%, and AUC of 42.46%, possibly due to insufficient modeling of inter-channel dependencies or short-term temporal dynamics. Specifically, WRFE module produced values 11% higher. In addition, WRFE module outperformed MuLHiTA (Xia et al., 2023) and BIOT (Yang et al., 2023) with F1 22% higher and 29% higher. It is likely that MuLHiTA focused on the long-term temporal dependencies of EEG signals while overlooking their spatial features. In contrast, BIOT concerned about features between channels. And it is worth mentioning that the Riemannian module used in WRFE module has yet to provide significant improvements. This may be because the Riemannian manifold, which is based on manifold structure, exhibits greater robustness in extracting features between channels.

## 3.3 ABLATION STUDY

To further dissect the contributions of individual components within our proposed modules, we conducted comprehensive ablation studies.

Table 2: Comparison between BDCD module and SOTAs

| Method | SWD ↓ | Context-FID ↓ |
|---|---|---|
| **MIST-CNU Dataset** | | |
| C-WGAN | 0.0961 ± 0.0269 | 11.7481 ± 0.5893 |
| TimeGAN | 0.0979 ± 0.0165 | 17.0576 ± 3.5632 |
| Diffusion-TS | 0.1188 ± 0.0175 | 1.3249 ± 0.3534 |
| **BDCD module** | **0.0828 ± 0.0254** | **0.8232 ± 0.1961** |
| **MIST-UTP Dataset** | | |
| C-WGAN | 0.1216 ± 0.0113 | 43.4855 ± 2.8523 |
| TimeGAN | 0.1187 ± 0.0124 | 17.8896 ± 2.9578 |
| Diffusion-TS | 0.0575 ± 0.0159 | 13.7015 ± 0.9915 |
| **BDCD module** | **0.0478 ± 0.0029** | **13.2383 ± 4.7763** |
| **Rest-Task Dataset** | | |
| C-WGAN | 0.1323 ± 0.0341 | 25.3735 ± 2.3581 |
| TimeGAN | 0.0977 ± 0.0498 | 10.5505 ± 1.0178 |
| Diffusion-TS | 0.1063 ± 0.0028 | 8.0249 ± 1.0740 |
| **BDCD module** | **0.0768 ± 0.0109** | **7.7617 ± 0.3611** |

Table 3: Comparison of SWD and Context-FID for different methods across datasets

| Method | SWD ↓ | Context-FID ↓ |
|---|---|---|
| **MIST-CNU dataset** | | |
| Diffusion-TS | 0.1188 ± 0.0175 | 1.3249 ± 0.3534 |
| Diffusion-TS + BDM | 0.0918 ± 0.0201 | 1.1303 ± 0.1360 |
| Diffusion-TS + MLoss | 0.0256 ± 0.0048 | 0.3743 ± 0.1528 |
| **BDCD module** | **0.0828 ± 0.0254** | **0.8232 ± 0.1961** |
| **MIST-UTP dataset** | | |
| Diffusion-TS | 0.0677 ± 0.0001 | 13.7015 ± 0.9915 |
| Diffusion-TS + BDM | 0.0577 ± 0.0284 | 12.1325 ± 0.0078 |
| Diffusion-TS + MLoss | 0.0581 ± 0.0245 | 12.0359 ± 1.7542 |
| **BDCD module** | **0.0454 ± 0.0012** | **11.4492 ± 2.6204** |
| **Rest-Task dataset** | | |
| Diffusion-TS | 0.1355 ± 0.0002 | 8.0249 ± 1.0740 |
| Diffusion-TS + BDM | 0.0952 ± 0.0012 | 7.8628 ± 1.8625 |
| Diffusion-TS + MLoss | 0.0958 ± 0.0020 | 7.8801 ± 3.0920 |
| **BDCD module** | **0.0725 ± 0.0001** | **6.7882 ± 0.1110** |

Table 4: Ablation study results for the WRFE module

| Method | ACC (%) | Pre (%) | AUC (%) | F1 (%) | Rec (%) |
|---|---|---|---|---|---|
| EEGNet | 35.57 ± 3.96 | 35.58 ± 3.78 | 56.23 ± 3.97 | 29.36 ± 2.54 | 35.57 ± 3.95 |
| EEGNet + RM | 40.21 ± 2.78 | 45.23 ± 2.71 | 66.83 ± 0.22 | 36.68 ± 0.46 | 40.22 ± 2.79 |
| EEGNet + WCov | 37.50 ± 1.71 | 42.91 ± 3.12 | 62.84 ± 2.45 | 31.03 ± 0.38 | 37.50 ± 1.71 |
| **WRFE** | **54.52 ± 0.13** | **67.01 ± 0.14** | **83.12 ± 0.04** | **51.12 ± 0.04** | **54.52 ± 0.01** |

### 3.3.1 ABLATION STUDY ON THE BDCD MODULE

An ablation study was conducted using the MIST-CNU, MIST-UTP, and Rest-Task datasets in Table 3. Starting from the Diffusion-TS baseline, we observe that adding either component improves performance. The integration of the BDM (Diffusion-TS+BDM) lowers the Context-FID, demonstrating that the task-guided attention mechanism plays a crucial role in steering the generation process towards more realistic task-state EEG patterns. Similarly, optimizing the model with our proposed MLoss (Diffusion-TS+MLoss) also leads to a significant drop in Context-FID, indicating its effectiveness in capturing the dynamic covariance features between EEG channels, a finding consistent with Trambaiolli et al. (Trambaiolli et al., 2020). Crucially, the full BDCD module, which combines both BDM and MLoss, achieves the best overall performance. This underscores the complementary nature of these two components: the BDM provides high-level semantic guidance from the task-state, while the MLoss ensures the fidelity of low-level statistical properties in the generated signals.

### 3.3.2 ABLATION STUDY ON THE WRFE MODULE

Using EEGNet as the baseline, we first added the Riemannian manifold (EEGNet+RM), which resulted in a 4.84% increase in accuracy. This improvement highlights the advantage of modeling the spatial covariance of EEG signals on a Riemannian manifold, which inherently respects the matrix structure of the data and avoids distortions common in Euclidean-based methods. Next, we incorporated wavelet convolution (EEGNet+WCov), leading to a 1.93% accuracy gain. This confirms the value of wavelet transforms in capturing multi-scale temporal features. The full WRFE module, which integrates both Riemannian features and wavelet convolution, achieves a remarkable 20% improvement in accuracy over the baseline. This significant gain demonstrates the powerful synergy between these two techniques. Wavelet convolution is highly sensitive to time-frequency variations, making it excellent for capturing transient temporal patterns, though it can be susceptible to noise. Conversely, Riemannian geometry provides a robust and stable representation of spatial relationships but may overlook rapid temporal changes.

### 3.4 VISUALISATION RESULT ANALYSIS OF BDCD MODULE

Our hypothesis that BDCD module can gradually extract latent features from resting-state EEG signals was tested using a visual evaluation, conducted by visualizing the EEG topographic maps during the denoising process of the BDCD module, as well as the attention weights from the bidirectional decoupling attention decoder. We randomly selected the generated data of the 21st subject and visualized the topographic map and attention weights after averaging. As shown in Figure 3, the cross-attention weights of the decoder in panel B gradually increase as the generation steps progress, indicating that BDCD module increasingly focuses on latent task-related features in rs-EEG signals. Panel C illustrates that neural activity in the bilateral temporo-parietal regions is significantly enhanced in the real task-state EEG, which is consistent with previous research findings. At the initial generation stage (S = 1), panel a represents the state, where task-related features have yet to be extracted. By S = 400, noticeable activations begin to emerge over the right hemisphere, particularly at electrodes TP8, P8, and P6. When S = 700, the left temporo-parietal regions, including TP7, P7, and PO7, also show distinct activation. The final predicted task-state EEG exhibits a spatial distribution pattern highly consistent with the real task state EEG, successfully uncovering the latent task-related features from the resting-state signals.

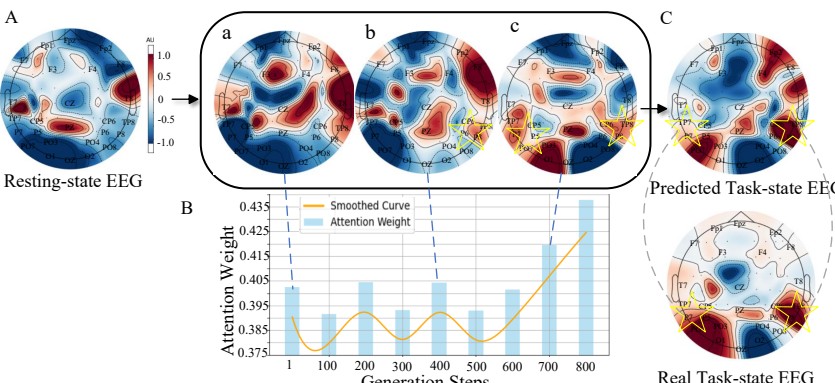

Figure 3: Visualization analysis process of BDCD module in extracting latent features from rs-EEG signals. A. The panel illustrates a step-by-step process by which BDCD gradually mines task-related latent features from rs-EEG signals and generates task-state EEG signals.

## 4 DISCUSSION AND LIMITATIONS

Limitation. Despite the significant progress achieved in this study, there remain areas for further improvement. Firstly, as this work is the first to propose the use of rs-EEG signals for identifying SD, there is a lack of mature baseline models and publicly available datasets for comparison. In future work, we plan to expand our current dataset and explore more reasonable and comprehensive methods for model evaluation. Secondly, the current model has not yet achieved full automation, and the process remains relatively complex. Show in Appendix.

## 5 CONCLUSION

In conclusion, ReTa-Diffusion presents a novel framework for early SD detection by uncovering latent task-state features within rs-EEG. Through diffusion modeling, attention fusion, and wavelet-Riemannian extraction, it achieves robust performance and offers new insights into EEG-based mental health assessment. The model was evaluated on three datasets: the private dataset MIST-CNU for four-class classification and the public dataset (Rest-Task dataset) for evaluating the BDCD module. ReTa-Diffusion provided the best results in terms of generalization ability and effectiveness compared to state-of-the-art frameworks (54.52% in Acc for MIST-CNU dataset). An ablation study demonstrated the effectiveness of each component in ReTa-Diffusion, thus effectively capturing latent features of rs-EEG signals and providing

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

# A APPENDIX

In this section, we provide a detailed description of our experimental paradigm and explain how the labels of restingstate EEG signals were obtained. The experimental dataset was collected in Montreal, where participants were induced into a task-state. The specific procedure is illustrated in Figure A1. The experiment consisted of two stages: the control stage and the stress stage. In the control stage, participants completed a Mental Arithmetic Task (MAT) without time constraints or verbal feedback. This stage aimed to familiarize participants with the task. In the stress stage, the procedure was similar to the control stage, but the MAT was timelimited to induce psychological stress. To minimize memory effects caused by learning during the first stage, there was an interval of at least seven days between the two stages. Each stage included an adaptation phase (5 minutes), a restingstate phase (5 minutes), and a psychological stress/control phase (28 minutes). The adaptation phase allowed participants to acclimate to the lab environment, eliminating potential environmental interference in data collection, and involved completing assessments such as the Adolescent Life Events Scale. During the resting-state phase, participants were asked to fixate on a "+" symbol displayed at the center of a blank screen, without performing any arithmetic tasks. The psychological stress/control phase involved the MAT, which was divided into four difficulty levels (L1–L4), with higher levels corresponding to greater computational difficulty. Finally, participants received feedback on their performance in the arithmetic tasks. In this study, we utilized the resting-state and task-state data from the psychological control stage.

To obtain the labels for resting-state EEG signals, we first recorded the scores of each participant on

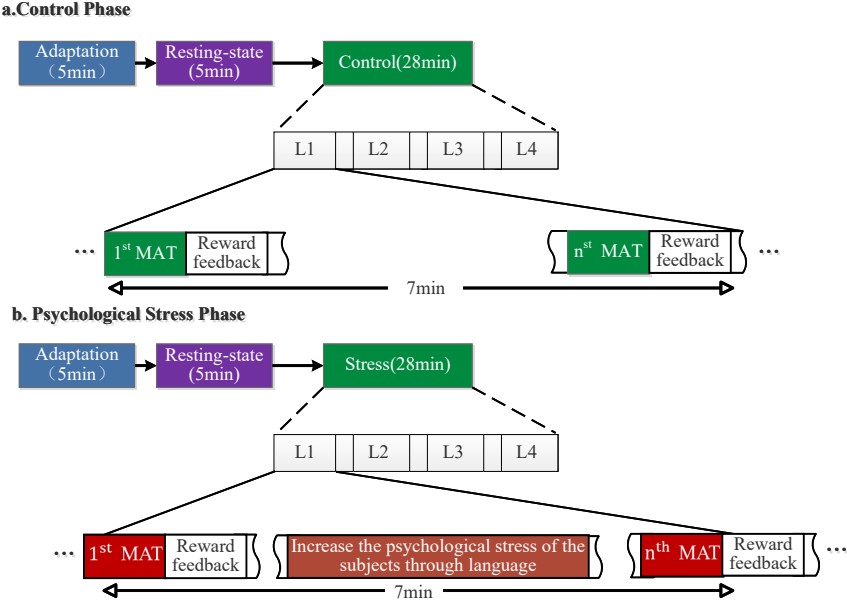

Figure A1: Experimental paradigm. The experiment included two stages: (a) a control phase and (b) a psychological stress phase. Each stage consisted of a 5-minute adaptation phase, a 5-minute resting-state phase, and a 28-minute task phase involving mental arithmetic tasks (MATs) of varying difficulty (L1–L4). In the stress phase, tasks were time-limited and verbal feedback was used to induce stress.

the Adolescent Life Events Scale. Then, following a commonly used quantile-based classification method in psychology, we categorized the stress levels of the resting-state EEG data. Specifically, we used the scores from 31 participants, sorted them in ascending order, and calculated the quartile thresholds based on Equation A1 to define different levels of psychological stress.

$$\hat{Q}(p) = x_k + (np - k)\left(x_{(k+1)} - x_k\right) \tag{A1}$$

Where $k = \lfloor np \rfloor$, $0 < p < 1$. We compute the quartile thresholds of the sample data for $p \in \{0.25, 0.50, 0.75\}$, thereby obtaining the sample distribution intervals, as shown in Table A1.

Table 5: Stress Levels and Participant Distribution (Table A1)

| Stress Level | Range (Score) | Number of Participants |
|---|---|---|
| Level 1 (L1) | 0–25 | 9 |
| Level 2 (L2) | 26–29 | 8 |
| Level 3 (L3) | 30–45 | 7 |
| Level 4 (L4) | > 45 | 7 |

## B    APPENDIX

The training pseudo-code of our ReTa-Diffusion is illustrated in Algorithm 1.

---
**Algorithm 1:** The Training pseudo-code for the ReTa-Diffusion

---
**Input:** Task-state EEG data $X_{\text{task}}$, Resting-state EEG data $X_{\text{rest}}$, labels $Y$, number of epochs $E$, batch size $B$

**Output:** Predicted labels $P$ of target rest-state EEG signals

1   **for** *epoch = 1 to $E$* **do**
2     **for** *iteration = 1 to $E/B$* **do**
3       Forward propagation by Eq. (1);
4       Disentangle features with Task-Resting Encoder by Eq. (2)–(4);
5       Calibrate the latent task-relevant features from resting-state EEG with bidirectional decoupling attention decoder by Eq. (4)–(10);
6       Capture temporal and spatial features by Eq. (11)–(17);
7       Apply fully connected layer and activation function to get $P$;

---

The testing pseudo-code of our ReTa-Diffusion is illustrated in Algorithm 2.

---
**Algorithm 2:** The Testing pseudo-code for the ReTa-Diffusion

---
**Input:** Resting-state EEG data $X_{\text{rest}}$, number of epochs $E$, batch size $B$

**Output:** Predicted labels $P$ of target rest-state EEG signals

1   **for** *iteration = 1 to $E/B$* **do**
2     Disentangle features with Task-Resting Encoder by Eq. (2)–(4);
3     Calibrate the latent task-relevant features from resting-state EEG with bidirectional decoupling attention decoder by Eq. (4)–(10);
4     Capture temporal and spatial features by Eq. (11)–(17);
5     Apply fully connected layer and activation function to get $P$;

---

## C    APPENDIX

You may include other additional sections here.

## D    APPENDIX

Appendix D: Case Study The complete procedure is illustrated in Figure D1. It includes the following stages. Figure D1: A case study on stress level detection using rs-EEG

D1. Experimental Design and Data Acquisition Five participants were recruited for this study. The experimental protocol commenced with each participant maintaining a five-minute period of quiet, eyes-closed rest. During this interval, resting-state Electroencephalography (EEG) signals were continuously recorded. Subsequent to acquisition, the raw data underwent preprocessing and were then input into our custom-developed stress recognition model.

D2. Model Processing Pipeline The model's processing pipeline consists of three core stages: EEG Signal Decoupling and Reconstruction: The data is first processed by the BDCD (Brain-State Decoupling and Reconstruction) module. The primary function of this module is to decouple the intrinsic correlations between resting-state and task-state EEG signals. Based on this analysis, it generates a "task-oriented resting-state EEG signal" that encapsulates information pertinent to the task.

Table 6: Subject-wise classification performance

| Subject | ACC (%) | F1 (%) | Predict Result | Label |
|---------|---------|--------|----------------|-------|
| Subject 1 | 100 | 100 | 1 level | 1 level |
| Subject 2 | 100 | 100 | 2 level | 2 level |
| Subject 3 | 44.8 | 61.88 | 4 level | 4 level |
| Subject 4 | 39.4 | 56.53 | 3 level | 3 level |
| **Mean** | **71.05** | **79.60** | - | - |

This stage is designed to extract latent, task-specific features from the resting-state data. Spatio-Temporal Feature Extraction from Generated EEG: The synthesized signal from the preceding stage is then fed into the WRFE (Wavelet-based Spatio-Temporal Feature Extraction) module. This module leverages wavelet transforms to extract key features associated with stress states from both the temporal and spatial dimensions of the signal. Feature Fusion and Stress Classification: The resulting spatio-temporal feature vectors are integrated via a fusion layer and subsequently passed to a classifier. The classifier performs the final discrimination of the participant's stress state, outputting a corresponding stress level (e.g., Level 1 - Low, Level 2 - Moderate, Level 3 - High). D3. Validation and Performance Evaluation To validate the model's predictive accuracy, its outputted stress levels were benchmarked against the results from standardized psychological questionnaires completed by the participants (see Appendix 3). The model's performance was quantitatively assessed and calibrated by calculating the concordance between its predictions and the questionnaire-based assessments. This analysis was supplemented with visualization charts to intuitively demonstrate the model's classification efficacy.

The results, shown in the table below, reveal a distinct performance pattern: the model excels at predicting low-stress states (levels 1 and 2) but shows reduced accuracy for high-stress states (levels 3 and 4). This phenomenon can be interpreted from a neurophysiological perspective. In low-stress states, brain activity is more regulated and differentiated. The covariance matrices of EEG signals exhibit pronounced spatial differences across brain regions, particularly in the prefrontal and parietal cortices, which are associated with executive function and emotional regulation. These distinct spatial patterns provide clear, discriminative features that the model can easily learn, leading to high prediction accuracy. In contrast, high-stress states often trigger a more homogenized and widespread brain response, characterized by high-arousal, less differentiated activity across multiple regions. This "neural saturation" effect reduces the distinctiveness of spatial features in the EEG covariance matrices, making it more challenging for the model to identify reliable patterns for accurate classification. Consequently, the model's performance declines for these higher stress levels. This finding not only highlights a current limitation of our model but also aligns with existing neuroscience literature, suggesting that the neural signatures of high stress are inherently more subtle and variable across individuals.

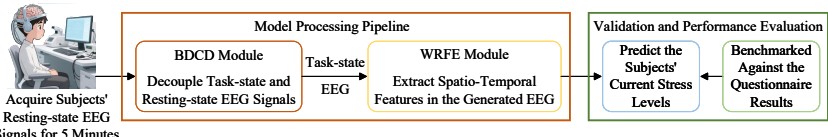

Figure A1: Experimental paradigm. The experiment included two stages: (a) a control phase and (b) a psychological stress phase. Each stage consisted of a 5-minute adaptation phase, a 5-minute resting-state phase, and a 28-minute task phase involving mental arithmetic tasks (MATs) of varying difficulty (L1–L4). In the stress phase, tasks were time-limited and verbal feedback was used to induce stress.

# E    APPENDIX

Three datasets are employed to evaluate the performance of ReTa-Diffusion for metal stress level detection, where both MIST-CNU and MIST-UTP are private datasets, and Rest-Task dataset is a

publicly available dataset.

**MIST-CNU** and **MIST-UTP** designed based on the Montreal Imaging Stress Task (MIST) paradigm, consisting of resting-state phase and task-state phase. The task-state phase included stress and control sessions, each of which involved one four-level (levels 1–4) mental arithmetic task with a duration of 20 min (4 levels × 5 min/level). MIST-CNU comprises EEG signals using 64 electrodes from 31 subjects (5 males and 26 females) with a sampling rate of 500 Hz, and the MIST-UTP includes 31 subjects (9 males and 22 females) recorded at 500 Hz using 20 electrodes.In particular, prior to EEG signal acquisition, we collected responses to the Adolescent Life Events Scale to generate labels $y \in \mathbb{R}^N$, where $N$ is the number of generation samples. Detailed information can be found in Appendix A.The network was evaluated using a fivefold CV, and the 31 subjects were divided into two groups: 27 for training and validation, and 4 for testing.

**Public Rest-Task dataset** utilized in this study is a EEG dataset across multiple subject-driven states. It includes recordings from resting states and subtraction tasks using 61 electrodes, where 21 participants (4 males and 17 females) were employed with a sampling rate of 500 Hz.

**System Environment and Parameter Configuration.** ReTa-Diffusion is implemented in a Python environment using Pytorch-cuda 12.1 backend on a NVIDIA GeForce RTX4090. Adam was adopted to optimize the network. The initial learning rate was set to 1e-6.

**Evaluation Metrics.** In order to better observe and evaluate the proposed method, we calculate the following evaluation metrics: sliced wasserstein distance (SWD)(Similarity evaluation), context-Frechet´ inception distance (Context-FID)(Similarity evaluation), accuracy (Acc), precision (Pre), recall (Rec), F1 score (F1), Area Under Curve (AUC).

In this study, we propose a framwork termed ReTa-Diffusion for early detection of SD using rs-EEG signals. Unlike existing studies that focus on evaluating SD using both rs-EEG and task-state EEG signals (Ghiasi et al. 2021; Earl et al. 2024; Xue et al. 2024), we emphasize the importance of identifying latent task-related neural activity within rs-EEG signals, providing new perspectives and approaches for early detection of SD. The proposed ReTa-Diffusion integrates two key modules: BDCD and WRFE. In the former, inspired by techniques in image style transfer (Zhang et al. 2023; Deng et al. 2022; Kwon et al. 2023), we have designed a bidirectional decoupling strategy to disentangle features from rs-EEG and task-state EEG signals. Considering the high noise levels and complex distributions inherent in EEG signals, we adopted a Diffusion-TS model (Yuan et al. 2024) as the backbone network to leverage its advantages in stable modeling and high-dimensional signal generation. To effectively explore the deep associations between rs-EEG and task-state EEG signals, we introduced a multi-layer attention mechanism. Previous work (Jiang et al. 2024) attempted to convert multi-channel EEG signals into images to facilitate the diffusion model's ability to capture spatial dependencies, in which the transformation often led to the loss of temporal information due to the compression of time-series data during image conversion. To address this limitation, we proposed a multi-dimensional sliding window loss function that preserved the temporal structure of EEG signals during training. The WRFE module specifically addresses the challenge of inadequate feature representation inherent in rs-EEG signals for the identification stage. By combining wavelet convolution with Riemannian manifold based feature extraction, it adeptly captures both transient temporal dynamics and functional connectivity within these signals. In contrast, previous studies have primarily relied on conventional time-domain or frequency-domain analysis methods, such as power spectral density (PSD) analysis (Liu et al. 2022). These approaches typically depend on global statistical features and often fall short in revealing the non-stationary dynamics that occur at short time scales. To overcome existing limitations, we adopt Riemannian manifold to transform EEG signals into covariance matrices and perform modeling on the Riemannian manifold, enabling more robust and invariant functional connectivity analysis (Paillard et al. 2025).

