# OpenReview forum: "ReTa-Diffusion: Exploring Task-State from Resting-State in EEG Signals via Bidirectional Decoupling and Latent Guiding for Early Detection of Subclinical Depression"
_ICLR.cc/2026/Conference — ICLR 2026 Conference Withdrawn Submission_

### Official Review · Reviewer_WD3Q · 2025-10-23

**Soundness:** 3
**Presentation:** 2
**Contribution:** 2
**Rating:** 4
**Confidence:** 4

**Summary:**

This paper introduces ReTa-Diffusion, a novel framework that predicts subclinical depression by uncovering task-like neural signatures hidden in resting-state EEG without requiring any actual task recordings. Its core innovation is a bidirectional conditional diffusion module that disentangles and aligns resting and task-state features to synthesize realistic task-state EEG signals from rs-EEG. These synthetic signals are then fed to a Wavelet–Riemannian feature extractor that captures both transient temporal dynamics and spatial covariance structure for stress-level classification. Extensive five-fold cross-subject validation on three datasets shows 54.52 % multiclass accuracy, outperforming state-of-the-art baselines by 18.95 %, while ablations and visualizations confirm that the model successfully mines latent task-related activity from rs-EEG and generalizes across subjects.

**Strengths:**

Innovative framework: Introduces ReTa-Diffusion, a pioneering framework for extracting task-related features from resting-state EEG  signals, enabling early detection of subclinical depression (SD). First work (to the authors’ knowledge) to generate task-state EEG from rs-EEG for SD detection.
Multimodal feature extraction: Combines wavelet convolution with Riemannian manifold techniques to capture dynamic temporal patterns and functional connectivity. Provides a robust and invariant feature representation, addressing challenges in rs-EEG signal analysis.

**Weaknesses:**

Complexity and Lack of Automation: The current model has not yet achieved full automation and operates as a non-end-to-end framework, requiring multiple sequential processing stages that remain relatively complex.
Limited interpretability: The paper does not explain what task-state EEG is being generated, or how it relates to depression-specific biomarkers.
Related work: A dedicated Related Work section should be added to systematically review prior contributions and explicitly articulate the incremental value and fundamental distinctions of this study.

**Questions:**

Low absolute accuracy: 54.52% on a 4-class task is only slightly better than random (25%). While it outperforms baselines, the clinical utility is questionable. Although the proposed method outperforms the baselines cited in the paper, how do the authors demonstrate that the research can actually be deployed and scaled in real-world practice?

---

### Official Review · Reviewer_ugVP · 2025-10-30

**Soundness:** 2
**Presentation:** 2
**Contribution:** 2
**Rating:** 2
**Confidence:** 4

**Summary:**

This paper introduces a novel framework named ReTa-Diffusion for the early detection of subclinical depression (SD) by analyzing resting-state EEG (rs-EEG) signals. The core idea is to leverage task-state EEG (task-EEG) as guidance to decouple and generate task-relevant features from rs-EEG. The framework consists of two main modules: a Bidirectional Decoupled Conditional Diffusion (BDCD) module to generate task-state EEG from rs-EEG, and a Wavelet-Riemannian Feature Extraction (WRFE) module to extract spatio-temporal features from the generated signals for classification. The authors conduct experiments on three datasets and report that their model achieves state-of-the-art performance on multiclass classification tasks.

**Strengths:**

1.  **Novel Problem Formulation and Potential Impact**: The core idea of the paper—inferring and generating task-informed EEG from easily acquirable rs-EEG for early SD screening—is highly original. If successful, this approach could significantly lower the barrier for large-scale mental health screening, offering a highly inspiring new direction for the field.
2.  **Integration of Advanced Techniques**: The study demonstrates the ability to integrate multiple state-of-the-art techniques (conditional diffusion models, bidirectional decoupled attention, wavelet transforms, and Riemannian geometry) into a complex and coherent framework. This multi-faceted technical combination is methodologically advanced and showcases the authors' strong technical proficiency.
3.  **Intuitive Visualization**: Figure 3 provides compelling visual evidence demonstrating how the model progressively generates neural activity patterns from rs-EEG that are highly similar to real task-EEG. This lends preliminary support to the model's core hypothesis and enhances its interpretability.

**Weaknesses:**

Despite its novel idea, this paper suffers from several major flaws in scientific rigor, experimental validation, and academic presentation, which severely undermine the reliability of its conclusions.

**1. [Major Flaw] Fundamental Mismatch Between Research Problem and Experimental Paradigm**
This is the most critical issue in the paper. The stated goal is to detect "Subclinical Depression" (SD), a chronic and persistent mood disorder. However, the experimental paradigm employs an "Acute Psychological Stress" task induced by time-limited mental arithmetic.
*   **Conflation of Concepts**: Depression and acute stress are fundamentally different in their neurophysiological mechanisms, time scales (chronic vs. transient), and underlying neural markers. The authors provide no scientific or clinical evidence to support the assumption that EEG features from acute stress can serve as reliable biomarkers for a chronic depressive state.
*   **Misleading Conclusions**: Generalizing conclusions from an acute stress model to depression screening is scientifically unsubstantiated. This makes the paper's core contribution built upon an unverified, and likely incorrect, premise, severely weakening the entire scientific value of the work.

**2. [Major Flaw] Critically Flawed Experimental Design and Evaluation**
The experimental framework is insufficient to support the paper's claims, and some results even contradict its own assertions.
*   **Lack of a Crucial Baseline**: The paper's central claim is the superiority of its two-stage (generation + classification) framework. However, it completely omits the most important and direct baseline: **training a strong classifier directly on the original rs-EEG data** (e.g., feeding rs-EEG directly into EEGNet or the proposed WRFE module). The authors' claim that an "end-to-end comparison is not feasible" is unconvincing. Without this baseline, it is impossible to determine whether the complexity introduced by ReTa-Diffusion is beneficial, or potentially even detrimental.
*   **Contradictory Ablation Study Results**: The results in Table 3 are perplexing. On the MIST-CNU dataset, "Diffusion-TS + MLoss" (Context-FID: 0.3743) performs significantly better than the full BDCD module (0.8232). This strongly suggests that the addition of the BDM (Bidirectional Decoupling Module) is actually **harmful** to the generation quality. The authors offer no explanation for this counter-intuitive finding. Furthermore, the paper does not clearly define what BDM and MLoss are, making the results difficult to interpret.
*   **Insufficient Validation of the Core Task**: The main classification performance (Table 1) is reported on only a single private dataset (MIST-CNU). For a method claimed to have broad applicability, the lack of validation on other datasets severely limits the generalizability and credibility of the findings.

**3. [Severe Flaw] Extremely Poor Academic Presentation and Reproducibility**
The paper's writing, structure, and attention to detail fall far short of the standards for a top-tier conference.
*   **Missing Core Section**: The paper **completely lacks a "Related Work" section**. This makes it impossible for reviewers to situate the work within the context of existing research and makes the authors' claims about the relationship between rs-EEG and task-EEG appear unsubstantiated.
*   **Absence of Critical Methodological Details**: The paper fails to explain a core implementation detail: how the 5-minute rs-EEG segments are **temporally aligned and paired** with the 28-minute, multi-level task-EEG data during training. This omission makes the method entirely irreproducible.
*   **Numerous Writing and Formatting Errors**: An overly long abstract, lack of paragraph spacing, an empty Appendix C, an unfinished conclusion section, incorrect table citations (Section 3.2.1 discusses generative models but cites Table 1 instead of Table 2), and inappropriate content in the supplementary materials (e.g., screenshots of chats in Chinese) all reflect a profound lack of rigor and care in manuscript preparation.
*   **Private Datasets**: Both datasets used for the core task evaluation are private, which hinders the community from verifying and building upon this work.

**Questions:**

1.  **Regarding the Scientific Hypothesis**: Can the authors provide evidence from neuroscience or clinical literature to support a strong correlation or transferability between EEG features induced by an acute mental arithmetic task and the neural markers of subclinical depression? If not, how do you justify the scientific validity of using an acute stress paradigm to address the problem of chronic depression detection?
2.  **Regarding the Critical Baseline**: Why was a direct end-to-end baseline, which involves training a classifier (e.g., EEGNet or your WRFE module) directly on the **original rs-EEG**, not included in the comparison? What makes such a comparison "not feasible"? Given that this is the most direct way to evaluate the added value of your entire framework, could you provide results for this experiment?
3.  **Regarding Method Reproducibility**: Could you please elaborate on the training details for the BDCD module? Specifically, how is a single 5-minute segment of rs-EEG **aligned or paired** with a 28-minute segment of task-EEG that contains four different stress levels? Is it a one-to-one mapping, one-to-many sampling, or another mechanism? What is the rationale behind this design choice?
4.  **Regarding the Contradictory Ablation Results**: In Table 3, the "Diffusion-TS + MLoss" configuration achieves a significantly better Context-FID score on the MIST-CNU dataset than the full BDCD model. Does this imply that the BDM component has a negative impact on generation quality? Please provide an explanation for this phenomenon.

---

### Official Review · Reviewer_GMLF · 2025-11-01

**Soundness:** 2
**Presentation:** 2
**Contribution:** 1
**Rating:** 4
**Confidence:** 4

**Summary:**

This paper introduces a novel framework for early detection of subclinical depression (SD) from resting-state EEG, leveraging a Bidirectional Decoupled Conditional Diffusion (BDCD) model to generate synthetic task-state EEG.

**Strengths:**

The motivation is well justified, and the method demonstrates promising results.

**Weaknesses:**

1. Overstated Claim of "Subclinical Depression" and Clinical Relevance

While the paper positions its objective as the early detection of subclinical depression (SD), the labels are derived from the Adolescent Life Events Scale, which captures self-reported stress exposure, not depressive symptoms or clinical assessments. Moreover, clinically oriented terms such as “diagnosis,” “early detection,” and “clinical screening” are used throughout, despite the lack of validated depression instruments and the small, non-clinical cohort. The terminology therefore overstates the clinical applicability of the work.

2. Limited Use of Public Dataset

Among the three datasets used, only one is publicly available. However, it is used solely for evaluating EEG generation quality with SWD and Context-FID. No classification results are reported on this dataset, limiting both reproducibility and external validation of the core claims.

3. Insufficient Justification for Diffusion Modeling
The use of a computationally expensive diffusion model is insufficiently justified. The paper does not provide comparative ablations or motivate why a diffusion-based approach is preferable over more straightforward generative models (e.g., autoencoders, domain adaptation methods) in this specific EEG application.

4. Missing Strong Baselines

The paper ignores comparisons to recent strong methods in EEG decoding. For example, CBraMod (ICLR 2025) offers open-source code and pretrained models, and achieves state-of-the-art results across multiple EEG decoding benchmarks.

Ref:

[1] Wang, Jiquan, et al. "CBraMod: A Criss-Cross Brain Foundation Model for EEG Decoding." The Thirteenth International Conference on Learning Representations.

**Questions:**

See comments above.

---

### Official Review · Reviewer_jTjj · 2025-11-01

**Soundness:** 1
**Presentation:** 2
**Contribution:** 2
**Rating:** 2
**Confidence:** 3

**Summary:**

This paper proposes ReTa-Diffusion, a novel framework for detecting Subclinical Depression (SD) from resting-state EEG (rs-EEG). The core idea is that rs-EEG, while easily acquired, lacks task-specific information, but contains the neural substrates of potential task-states. The framework has two stages: 1) A Bidirectional Decoupled Conditional Diffusion (BDCD) module, a generative model to "translate" rs-EEG into corresponding, more informative task-state EEG. 2) A Wavelet-Riemannian Feature Extraction (WRFE) module that classifies the generated task-state EEG using Gabor wavelets (for temporal features) and Riemannian geometry. The authors report a 54.52% accuracy on a 4-class classification task on the private MIST-CNU dataset, claiming to outperform SOTA methods.

**Strengths:**

The concept of using a generative model to translate from resting-state to task-state EEG is novel and theoretically interesting.
Tables 3 and 4 provide good internal validation, showing that the components of BDCD and WRFE are complementary and that their combination (synergy) leads to the best performance within the paper's experimental setup.
Figure 3 provides strong qualitative evidence for the BDCD module, showing it learns to generate neuro-physiologically plausible task-state patterns.

**Weaknesses:**

The model fails to detect the target high-stress classes (L3/L4), as shown in Appendix D, Table 6. This invalidates the paper's core claim.
The paper fails to cite the foundational work on Riemannian EEG classification (Barachant et al.)  and fails o discuss a highly similar, contemporaneous method (Paillard et al., GREEN) , undermining the novelty of the WRFE module.
 Key SOTA baselines (BIOT, LaBraM)  perform at random-chance levels , making the claimed 18.95% improvement not credible.
The 54.52% average accuracy  is both clinically unusable for a 4-class problem and a misrepresentation of the true (and failed) performance on the target classes.

**Questions:**

see weakness

---

### Note · Authors · 2025-11-19

I have read and agree with the venue's withdrawal policy on behalf of myself and my co-authors.